# Morpho-Cultural and Pathogenic Variability of *Sclerotinia sclerotiorum* Causing White Mold of Common Beans in Temperate Climate

**DOI:** 10.3390/jof8070755

**Published:** 2022-07-21

**Authors:** Roaf Ahmad Rather, Farooq Ahmad Ahanger, Shafat Ahmad Ahanger, Umer Basu, M. Altaf Wani, Zahida Rashid, Parvaze Ahmad Sofi, Vishal Singh, Kounser Javeed, Alaa Baazeem, Saqer S. Alotaibi, Owais Ali Wani, Jasima Ali Khanday, Showket Ahmad Dar, Muntazir Mushtaq

**Affiliations:** 1Division of Plant Pathology, Faculty of Agriculture, Sher-e-Kashmir University of Agricultural Sciences and Technology of Kashmir, Wadura, Sopore 193201, India; ratherrouf99@gmail.com (R.A.R.); shafatahanger99@gmail.com (S.A.A.); jasimaalik@gmail.com (J.A.K.); 2Krishi Vigyan Kendra, Sher-e-Kashmir University of Agricultural Sciences and Technology of Kashmir, Shuhama, Ganderbal 190006, India; 3Division of Entomology, Indian Agricultural Research Institute, New Delhi 110012, India; basuumar1608@gmail.com; 4Division of Genetics and Plant Breeding, Faculty of Agriculture, Sher-e-Kashmir University of Agricultural Sciences and Technology of Kashmir, Wadura, Sopore 193201, India; wani.altaf100@gmail.com (M.A.W.); parvazesofi@gmail.com (P.A.S.); 5Dry Land Agriculture Research Station, Rangreth, SKUAST-K, Shalimar 190025, India; zahida1926@gmail.com; 6Genetics and Plant Breeding, Dr. Rammanohar Lohia Avadh University, Ayodhya 224123, India; vishalsinght72@gmail.com; 7Division of Fruit Science, Faculty of Agriculture, Sher-e-Kashmir University of Agricultural Sciences and Technology of Kashmir, Wadura, Sopore 193201, India; kousarjavaid11@gmail.com; 8Department of Biology, College of Science, Taif University, P.O. Box 11099, Taif 21944, Saudi Arabia; aabaazeem@tu.edu.sa; 9Department of Biotechnology, College of Science, Taif University, P.O. Box 11099, Taif 21944, Saudi Arabia; saqer@tu.edu.sa; 10Division of Soil Science and Agricultural Chemistry, Faculty of Agriculture, Sher-e-Kashmir University of Agricultural Sciences and Technology of Kashmir, Wadura, Sopore 193201, India; owaisaliwani@gmail.com; 11Faculty of Forestry, Sher-e-Kashmir University of Agricultural Sciences and Technology of Kashmir, Benhama, Ganderbal 191202, India; drshowketshameem@gmail.com; 12School of Biotechnology, Sher-e-Kashmir University of Agricultural Sciences and Technology, Chatha, Jammu 180009, India

**Keywords:** common bean, germplasm, mycelial compatibility groups, pathogenic variability, *Sclerotinia sclerotiorum*, white mold

## Abstract

The present systematic research on cultural, morphological, and pathogenic variability was carried out on eighty isolates of *Sclerotinia sclerotiorum* collected from major common bean production belts of North Kashmir. The isolates were found to vary in both cultural and morphological characteristics such as colony color and type, colony diameter, number of days for sclerotia initiation, sclerotia number per plate, sclerotial weight, and size. The colony color ranged between white and off-white with the majority. The colony was of three types, in majority smooth, some fluffy, and a few fluffy-at-center-only. Colony diameter ranged between 15.33 mm and 29 mm after 24 h of incubation. The isolates took 4 to 7 days for initiation of sclerotia and varied in size, weight, and number per plate ranging between 14 and 51.3. The sclerotial arrangement pattern on plates was peripheral, sub peripheral, peripheral, and subperipheral, arranged at the rim and scattered. A total of 22 Mycelial compatibility groups (MCGs) were formed with seven groups constituted by a single isolate. The isolates within MCGs were mostly at par with each other. The six isolates representing six MCGs showed variability in pathogenicity with isolate G04 as the most and B01 as the least virulent. The colony diameter and disease scores were positively correlated. Sclerotia were observed to germinate both myceliogenically and carpogenically under natural temperate conditions of Kashmir. Germplasm screening revealed a single resistant line and eleven partially resistant lines against most virulent isolates.

## 1. Introduction

The common bean (*Phaseolus vulgaris* L.), a *Leguminosae* crop, is the most widely grown legume in the world including India, where it is locally known as Rajmash [1]. The crop is suitable for a wide range of agro-ecosystems including tropical, sub-tropical, and temperate regions and thus is distributed worldwide [2]. In India, French beans rank second to peas in production and are extensively grown in Himachal Pradesh, Uttar Pradesh, Uttrakhand, Karnataka, Maharashtra, and Jammu and Kashmir [3]. Despite significant increases in productivity and output, a big gap continues between yield potential of the released varieties and yield achieved in the farmer’s field, mostly due to abiotic and biotic stresses. The common bean succumbs to various destructive diseases caused by plant pathogenic bacteria, fungi, and viruses. Amongst them, the white mold of beans, also called sclerotinia rot or watery pod rot of the common bean, caused by *Sclerotinia sclerotiorum* (Lib.) de Bary, is one of the most devastating which limits crop productivity and curtails the exchange value of produce [4,5].

The white mold disease of the bean warrants immediate intervention since it causes substantial yield losses to the produce and in severe infection at pod formation can exceed up to 100 per cent, obstructing the profitable cultivation of the crop [6]. The disease symptoms are frequently observed on all plant parts including leaves, stems, and pods. Infection generally starts around 10–15 cm above the soil level approximately near the stem-petiole junction. The dark-brown water-soaked spots begin first on the infected leaves and petioles and spread promptly to the stem and branches. During the later stages of disease, superficial cottony mycelial growth of the pathogen is observed on the infected stem, petiole, and pods. As the disease progresses, the pods exhibit brown discoloration and soft rot, resulting in death of infected branches. The fungus produces resting spores (sclerotia) at this stage, which are initially white and then turn black with maturity [7]. Mature black sclerotia are released into the soil during harvesting and get mixed with the harvested seed and then serve as the primary source of inoculum for the next year [8]. Sclerotia contribute a significant role as the inoculum source and as main structures for long-term survival of the pathogen. For germination (myceliogenically or carpogenically), the sclerotia necessitate prolonged periods of moist soil and a moderate temperature [9]. The sclerotia are capable of surviving in the soil for up to eight years [10]. Carpogenic germination of sclerotia releases ascospores which are transmitted by wind to distant places and act as a primary source of inoculum [11].

In Kashmir, this disease has assumed an alarming threat in low lying areas [12]. Various fungicides, bio-control agents, and plant extracts are used to manage white mold disease [7,13]. However, increasing consciousness about the development of phytopathogenic fungicide-resistant strains and adverse consequences of pesticides on the soil ecosystem and on plant, animal, and human health and accumulation of high levels of pesticide residues in the edible parts have compelled phyto-pathologists to look for alternative ecofriendly strategies for management of plant diseases [14,15]. Further, the soil and seed-borne nature of the disease renders complete disease control through fungicidal use a difficult task [15,16]. Moreover, the vigorous nature of the pathogen leads to heavy economic losses in only few days of optimum environmental conditions for the disease development. Further, no report has been found available regarding the screening of the bean germplasm for resistance against this disease from India. Moreover, very few reports of resistant germplasm are available at the international level [17].

Hence, there is promise of screening the common bean germplasm and resistance source; if any are found, they can be used in breeding programs to develop resistant varieties against white mold disease. Planting white mold resistant varieties seems to be a very attractive and reliable alternative for disease management. To screen the germplasm for resistance against the pathogen, understanding the population structure of the pathogen is of foremost importance. Hence, the present study was devised to know the morpho-cultural characteristics and pathogenic variability of the pathogen in Kashmir and screen the available bean germplasm against the mold pathogen.

## 2. Materials and Methods

The study was conducted in the laboratory and greenhouse, Division of Plant Pathology, Faculty of Agriculture, Sher-e-Kashmir University of Agricultural Sciences and Technology of Kashmir, Wadura, Sopore, J&K, India. The samples for the study were obtained during the survey for the disease status from North Kashmir.

### 2.1. Collection of Diseased Plant Material

Infected bean plants showing typical symptoms of white mold including water-soaked lesions, white cottony fungal mycelial growth, and black sclerotia on leaves, petiole, stem, etc. were collected in paper bags from the bean fields of different districts of North Kashmir and were brought to the laboratory for further investigation.

### 2.2. Isolation and Purification of Pathogen

The collected diseased specimens in the laboratory were washed with running tap water to remove the adherent dirt particles and then rinsed with distilled water. The infected plant parts were then cut into small bits. The bits and sclerotia were surface sterilized with 0.1 per cent mercuric chloride (HgCl_2_) solution for 30 s and rinsed three times in sterile distilled water. After that, the pieces and sclerotia were blotted dry, transferred aseptically to a potato dextrose agar (PDA) medium under a laminar air flow cabinet and incubated at 23 ± 2 °C. After 3–6 days of incubation, the mycelium/sclerotia formed were shifted to freshly prepared PDA slants/petri-dishes and incubated at 23 ± 2 °C for 7 days. The pure culture obtained was sub-cultured after every month and stored at 4 °C in the refrigerator for further studies. A total of 80 isolates of the white mold pathogen *S. sclerotiorum* (Lib) de Bary were obtained from sclerotia-diseased samples collected from geographically diverse parts of North Kashmir. The isolates were grouped and designated according to the geographical location. Eight isolates (N01–N08) were obtained from the district of Bandipora, seventeen (G01–G17) from the district of Ganderbal, nineteen (K01–K19) from the district of Kupwara, and thirty-six (B01–B36) from the district of Baramulla (Table 1). The number of isolates from each district is proportional to the area under the bean crop and the incidence of the disease.

### 2.3. Identification of the Pathogen

Various morphological and cultural characteristics comprising shape, size, color of fungal colony, mycelial growth rate, sclerotial characteristics, and symptoms produced on the host (bean plant) were used for the identification of the pathogen.

### 2.4. Mycelial Compatibility Grouping

*Sclerotinia sclerotiorum* is a homothallic fungus, and, when two heterothallic isolates are grown in the vicinity of each other in the same petri plate, a barrage zone or a zone of clearance is formed at the site of interaction of the colonies, termed as an incompatible reaction. The interaction is termed compatible when the two interacting isolates are homothallic and when the two colonies grow into each other without formation of a zone of clearance between them. However, the interaction is not visible clearly every time, and, to enhance the visibility of the interaction zone, red food coloring was used which leads to formation of a red line between the incompatible isolates visible at the rare side of the petri-dish, and no red line formation indicates the isolates to be compatible. To perform the mycelial compatibility test, mycelial discs (5 mm dia.) were cut from the margins of an actively developing colony of each isolate and placed triangularly on sterilized solidified Potato Dextrose Agar medium plates amended with red food coloring dye and incubated at 24 ± 20 °C (29). After about a week of incubation, the plates were observed for the interaction zone. When, between the confronting paired isolates, an evident line of a mycelia-free zone or a barrage zone or demarcation zone or a red line on the rare part of the petri-dish is observed, the response was considered as incompatible, or as a compatible response when the demarcation line was absent between the paired of confronting isolates [18].

### 2.5. Morpho-Cultural Variability

Mycelial discs of 5 mm diameter of each isolate of *S. sclerotiorum* were taken from a 2–3-day-old actively growing colony and transplanted to fresh PDA medium containing petri-plates (90 mm dia.). The morpho-cultural characteristics of each isolate, specifically, colony diameter (24 and 48 h after incubation at 23 ± 2 °C), type, and color; number of sclerotia per plate; days to initiation of sclerotia; weight; size; and pattern of formation after 15 Days After Incubation (DAI) were recorded to determine the morpho-cultural variability among the isolates. The sclerotial length and width (mm) were measured with the aid of a digital Vernier caliper, and three replications were used for each isolate.

### 2.6. Pathogenic Variability

The pathogenicity test of the *Sclerotinia sclerotiorum* isolates was determined by proving Koch’s postulates using the detached leaf and pod technique [12]. To evaluate pathogenic variability, from eighty isolates, forming twenty-two mycelial compatibility groups (MCGs), six MCGs were selected on the basis of the number of isolates in the MCG and the geographic distribution of the particular MCG, and, from each selected MCG, one isolate was selected on the basis growth characteristics. Since standard differentials for pathogenic variability in the bean crop are not available, eighteen genotypes of the bean were selected on the basis of best phenotypic and genotypic characteristics to differentiate pathogenic variability among isolates of *S. sclerotiorum*. Each genotype was sown in five pots with four plants maintained in each pot and was inoculated 3–5 weeks after sowing following the methods of Petzoldt and Dickson (1996) and Kull et al. (2004) [19,20]. Mycelial plugs of 5 mm were cut 1 cm back from the advancing margin of mycelial growth on 48-hr-old PDA culture maintained in the dark at 23 ± 2 °C. These mycelial plugs were placed fungus-side down, using a plastic straw on the cut stem or branch, gently pressed to ensure good contact. All inoculated pots were incubated in a mist chamber with the relative humidity maintained above 80 per cent and a temperature of 23 ± 2 °C.

### 2.7. Screening of Bean Germplasm

The experiments were carried out under greenhouse conditions in pots. Seeds of these genotypes were obtained from the Department of Genetics and Plant Breeding, FoA, Wadura, SKUAST-Kashmir. Under green/screen house conditions, seeds of all 63 genotypes were sown in pots filled with sterilized soil. A complete randomized design (CRD) was utilized with three replications, and three plants per replication of each genotype were maintained. The pathogenicity assessment was made using a resistance scale (Teran et al., 2006) (Table 1). The straw test scale used for the screening of the bean germplasm under greenhouse conditions is given in Figure 5.

### 2.8. Carpogenic Germination

Carpogenic germination of sclerotia of *S. sclerotiorum* is not reported *hit-here-to* in Kashmir, and primary infection is believed to be caused only by myceliogenic germination of sclerotia. However, a study was conducted to evaluate carpogenic germination of the sclerotia and pathogenicity of the resulting ascospores. Sclerotia were harvested from pure cultures on PDA produced in the lab in November and put in three beakers containing sterilized sand, kept outside to expose them to the natural climatic conditions, and were again brought to the lab in March for further studies. These beakers were regularly watered to keep them moist.

### 2.9. Data Analysis

The data collected during the study were statistically analyzed using different statistical techniques. The OPSTAT software package for agricultural research was used for the CRD one and two factor analysis, and a correlation analysis was performed using MINITAB and R software.

## 3. Results

The pathogen was isolated from the diseased bean plant parts showing distinctive white mold symptoms and produced typical sclerotia of *S. sclerotiorum*. A total of eighty isolates of the pathogen were prepared from the samples collected from the study area. The isolates were collected from four North Kashmir districts, specifically, thirty-six (B01–B36) isolates from the district of Baramulla, eight (N01–N08) from Bandipora, seventeen (G01–G17) from Ganderbal, and nineteen (K01–K19) from Kupwara. These cultures were maintained on PDA medium and stored as mycelial cultures and sclerotia at 4 °C in the refrigerator for further studies. The place of collection with topographic data and the study area map is given in Figure 1 and Table 2.

### 3.1. Mycelial Compatibility Grouping

Since the bean white mold pathogen, *S. sclerotiorum*, is a homothallic fungus during the mycelial compatibility test, the incompatible heterothallic isolates when put in a single petri dish develop an aversion zone at the line of contact between the two colonies after 3–6 days of growth due to the lysis of the incompatible fungal hyphae of the heterothallic colonies. When the genetically similar or homothallic colonies are put together in a petri dish, the colonies grow together without forming any clearance zone between them, and hence the interaction is termed compatible. However, the compatibility reaction results are not clearly visible always, and, in such cases, the medium needs to be amended by a food coloring agent, red coloring, to discern easily between the compatible and incompatible isolates and sometimes even requires to be observed under a microscope (Figure 2). Based on the mycelial compatibility test, the eighty *S. sclerotiorum* isolates of this study (Table 3) were assigned into 22 mycelial compatibility groups (MCGs) (Table 2). The MCGs were numbered in the order of the number of isolates in the group, with MCG-1 given to the MCG with the highest number of isolates, MCG-2 to the next highest, and so on. MCG-1 with the maximum number of isolates contained eleven isolates, followed by eight isolates in MCG-2, and then by six in both MCG-3 and MCG-4. MCGs 5, 6, 7, and 8 contained five isolates each, while MCGs 9, 10, 11, 12, and 13 contained three isolates each. MCGs 14 and 15 contained two isolates each, while MCGs 16, 17, 18, 19, 20, 21, and 22 contained a single isolate each.

The isolates from the adjoining geographical areas were likely to go into the same MCG; however, isolates were also observed to pair with isolates from distant locations. MCG-1 was formed by six isolates from the district of Baramulla and five from Kupwara. This MCG is present in the areas of Wadura, Nowpora, Brath, Armpora, and Hibdangerpora and in the Palhallan area of the district of Baramulla and adjoining areas of the district of Kupwara, namely, Sonwani, Unisoo, and Chougal. MCGs 2, 5, 9, and 14 consisted of isolates exclusively from Baramulla. MCG-2 is spread over the areas of Zaingeer, Sopore, and the Pattan areas of the Baramulla district which are adjacent to each other. Similarly, MCGs 11, 12, and 15 consisted of isolates exclusively from Kupwara and MCGs 4 and 10 from Ganderbal. MCG-3 consisted of two isolates from Ganderbal and four from Bandipora, while MCG-6 had four isolates from Baramulla and one from Bandipora, and MCG-7 had three isolates from Baramulla and two from Kupwara. MCG-8 consisted of isolates from three locations with two from Baramulla, one from Kupwara, and two from Bandipora. MCG-13 had two isolates from Ganderbal and one from Bandipora.

### 3.2. Morpho-Cultural Variability

Table 4 shows the variability in cultural characteristics of the eighty isolates. The isolates showed variations in their colony color, colony type, and mycelial growth rate. Most of the isolates produced white colonies and others produced a colony of off-white coloration. Six isolates, specifically, B06, B15, B20, B23, and B24 out of 36 isolates from Baramulla, produced an off-white colony on PDA plates, while from Bandipora all eight isolates produced a whitish colony. From Ganderbal, three isolates, G05, G08, and G13, produced an off-white colony, and from Kupwara, K03, K06, K07, K11, and K15, produced an off-white colony, while the rest showed whitish colonies (Figure 3).

Three types of colonies were observed, specifically, smooth, fluffy, and fluffy-at-center-only, with most isolates producing smooth colonies followed by fluffy. The isolates varied significantly in terms of colony diameter after 24 and 48 h of incubation (HrI). Colony diameter after 24 HrI was found to range from 15.33 mm in isolate G08 to 29.00 mm in isolate G04. The majority of isolates produced a 20 to 25 mm colony diameter. Similarly, the colony diameter after 48 HrI was observed to range from a mere 39.33 mm in isolate G08 to 69.67 mm in isolate G11. The isolates also showed variations in morphological characteristics such as the pattern of sclerotial formation, sclerotia size (length and width), sclerotial weight (fresh and dry), sclerotia number per 90 mm petri-plate containing PDA medium, and number of days to the initiation of sclerotia formation (Table 4). The color of sclerotia was black in all the isolates. The faster growing isolates took four to five days to initiate sclerotia formation, while the slower growing took six to seven days. The numbers of sclerotia per plate ranged from as low as 14 in isolate G08 and G14 to as high as 51.33 per plate in isolate B04. Most of the isolates produced 20 to 30 sclerotia per plate. The length ranged from 2.98 mm in isolate K19 to 10.86 mm in isolate B20. Similarly, the width ranged from 0.51 mm in isolate K17 to 4.45 mm in isolate G08. The isolates having longer sclerotia had lower widths. Fresh weight was taken after fifteen days of incubation, while dry weight was taken after subsequent drying. Isolate B25 had the maximum fresh weight of 1.45 g per plate, whereas isolate K19 had the lowest fresh weight of 2.98 g per plate. The dry weight was highest for isolate B20 at 0.51 g per plate.

The sclerotial arrangement was found to form five patterns: peripheral (P), sub-peripheral (SP), peripheral and sub-peripheral (P and SP), arranged at rim (AR), and scattered (SC) throughout the plate. Peripheral arrangement was seen in 36 isolates, sub-peripheral in 15, 13 with sclerotia arranged at rim, scattered, and peripheral, and sub-peripheral was found in 8 isolates each.

Principal component analysis (PCA) of different colony characteristics, specifically, color, type, initiation, and diameter varied significantly with size, weight, and pattern (Figure 4a,b). Each arm indicates the parameter direction and sides highlighted orthogonality, which were clustered in various groups. A PCA biplot was created (Figure 4a) in which two of the most contributing parameters are highlighted. However, in datasets where the correlation is not viable, we opt for dimensional reduction. The pairplot indicated the extent of the correlation in different colony parameters via scatter plot and histograms (Figure 4c). Further, multiple linear regression was estimated among the colony diameter with other colony characteristics of the pathogen. The relationship indicates a good correlation; however, minor dependence was observed among certain parameters, specifically, diameter and pattern, which explain variation among the isolates (Figure 4d).

### 3.3. Variability between MCGs

The twenty-two MCGs of this study were observed to show variation in terms of the average value of the parameters used to study the morphological and cultural variability. Table 5 shows the mean and critical difference (CD) values for different parameters of the twenty-two MCGs. The mean colony diameter was highest in MCG-3 with 27.33 mm after 24 HrI and 66.06 after 48 HrI and lowest for MCG-14 with 16.83 mm after 24 HrI and 44.16 mm after 48 HrI. MCG-3 was at par with MCGs 5, 20, and 18 and significantly higher than the rest in terms of colony diameter after 24 HrI. However, after 48 HrI, it was at par with MCGs 1, 5, 18, and 20 and significantly higher than the rest. The lowest colony diameter was seen in MCG-14 which was at par with MCGs 12 and 20 and significantly lowers than the rest. There was good variation among the MCGs in terms of the mean number of sclerotia per plate after 15 DAI. The highest number was found for MCG-18 with 42.33 sclerotia per plate and was significantly higher than the rest. However, this MCG consisted of a single isolate. Among the MCGs with two or more isolates, MCG-5 had the greatest mean number of sclerotia with 36.53 sclerotia per plate at 15 DAI and was at par with MCGs 3, 9, 11, 13, 14, 15, and 17. The lowest number of sclerotia at 15 DAI was found in MCG 19 with 16.67 sclerotia per plate, at par with MCGs 10, 16, 20, and 22 and significantly lowers than the rest.

The mean fresh weight ranged from 0.70 g in MCG-19 which was at par with MCGs 10, 12 17, and 20 and significantly lower than the rest; 1.25 g was observed in MCG-16 which was at par with MCGs 7 and 18 and significantly higher than the rest. The MCGs showed lesser variation in terms of the dry weight of sclerotia per plate. The highest dry weight was in MCGs 16 and 17 with 0.42 g, and, among MCGs with two or more isolates, the highest dry weight of 0.37 per plate was recorded for MCG-5. The lowest dry weight was recorded in MCG-12 with 0.22 g dry weight of sclerotia per plate at 15 DAI.

MCG-16 was found to form sclerotia of a maximum length of 7.6 mm which was at par with that of MCG-13 and MCG-6 and significantly higher than the rest of the MCGs. The least length was observed in MCG-22 with 2.98 mm length of sclerotia and was at par with MCG-10 and significantly shorter than the rest of MCGs. The MCGs showed lesser variation among themselves in terms of widths of sclerotia. The highest width of sclerotia was found in MCG-19 and was significantly higher than the rest, and the least width of sclerotia in MCG-11 was 2.05 mm wide, at par with MCGs 5, 7, and 13 and significantly lower than the rest.

### 3.4. Pathogenic Variation

To evaluate pathogenic variability, six isolates were selected from six MCGs on the basis of the number of isolates in the selected MCGs, the mean growth rate, and the area represented by the particular isolate. The reaction pattern of these six isolates of *S. sclerotiorum* on a set of eighteen putative differential bean lines was recorded using the scale of Teran et al. (2006) (Figure 5) [21] and is presented in Table 6 (Appendix A and Figure 6). The eighteen bean genotypes consisted of three released varieties, specifically, Shalimar French Bean-1, Shalimar Rajmash-1, and Arka-Anoop. The isolates varied in their virulence as is evident by the overall average disease reactions and the disease reaction by each isolate in each line. In the SFB-1 variety, the B01 isolate showed a significantly low disease score of 5.33 compared to other isolates which showed disease scores of 8 to 9. The isolates G11 and K01 showed the highest disease score of 9 in this variety and were significantly higher than isolate B01 and at par with other isolates. Similarly, in the SR-1 variety, isolates N07 and K01 produced the lowest disease score of 2.00, while the highest disease score of 5.33 was produced by isolate G04 and was significantly higher than the rest of isolates. In the Arka-Anoop variety, isolates G11, G04, and K01 produced the highest disease scores of 8.67 each, while isolate B01 produced the lowest score of 5.33. The overall mean disease score by the isolates was highest for isolate G04 followed by G11, and their scores were significantly higher than the rest. The isolate B05 was at par with isolates N07 and K01 and significantly higher than isolate B01 which was the least virulent isolate. The bean genotypes also varied in terms of disease scores. The most susceptible was SFB-1 followed by the Arka-Anoop variety, then by WB-923 and WB-642. Most of the lines were intermediate in their reaction to this pathogen with the intermediate disease scores varying between 4.11 in WB-112 and 6.56 in WB-184. Among the genotypes, only two genotypes, SR-1 and WB-1402, showed overall resistant disease scores against the mold pathogen.

### 3.5. Screening of Germplasm against White Mold of Bean

The most virulent isolate among the isolates evaluated for pathogenic variability “G04” was inoculated on sixty-three bean genotypes, and the results are shown in Figure 7; most of the genotypes succumbed to this isolate (Table 7). Among the 63 genotypes, 11 showed an intermediate reaction, and 1 genotype, WB-1402, was found to show a resistant reaction against the most virulent isolate. Among the three varieties, Shalimar Rajmash-1 (SR-1) showed an intermediate reaction, while the other two varieties, Arka Anoop and Shalimar French Bean-1 (SFB-1), were susceptible. The partially resistant lines included WB-112, WB-1607, WB-1006, WB-1118, WB-1634, WB-1446, WB-969, WB-54, WB-1441, WB-1643, and SR-1.

### 3.6. Carpogenic Germination

*Sclerotinia sclerotiorum* is a homothallic fungus and produces sclerotia as the survival, multiplication structures. These sclerotia germinate myceliogenically to produce vegetative hyphae causing infection directly or carpogenically to produce ascospores which then land on the host to produce disease. In Kashmir, however, the carpogenic germination was *hit-here-to* not reported, and only myceliogenic germination was presumed to initiate the infection, and sclerotia were believed to spread the inoculum. However, in this study, a lot of variability in terms of the geographical spread of MCGs was seen which suggested spread of the inoculum to longer distances through ascospores, and hence a study was undertaken to ascertain the involvement of sexual spores in spreading the disease. Sclerotia were harvested from pure cultures on PDA produced in the lab and put in three beakers containing sterilized sand, kept outside to expose them to the natural climate during the off-season (winter), and were again brought to the laboratory at the start of cropping season (spring). The sclerotia started germination in the third week of March by budding small eruptions which elongated to form stipes. The stipes were dark colored up to the point of submergence and light yellow above the level of water/sand. The stipes were pointed initially and showed phototrophic growth and varied from 10 to 50 mm in length, followed by formation of a cup-shaped structure (the apothecium) at the tip of the stipe, which later expanded to form a light yellowish, brown colored concave disc on which asci were organized and varied from 5 to 14 mm in diameter with the average being 7 mm. Asci were cylindrical in shape with a hyaline color and produced a closed packed mass with filiform paraphyses at the upper surface of the apothecium. Ascospores were one celled, elliptical to oval, hyaline, binucleate, and numbered eight in each ascus. The ascospores measured 9.44 to 12.72 µm (micrometer) in length and 4.06 to 6.42 µm in width (Figure 8). The first fully expanded apothecia were seen in the first week of April. By the third week of May, the majority of the sclerotia had produced apothecia. The ascospores were inoculated on the bean leaves under in vitro conditions and were observed to produce typical white-mold symptoms and sclerotia thereby proving their pathogenicity.

## 4. Discussion

The common bean (*Phaseolus vulgaris* L.) is one of the most important leguminous crops cultivated around the globe [22]. However, the bean crop is frequently affected by white mold caused by *S. sclerotiorum* (Lib.) de Bary resulting in serious yield reduction and post-harvest losses during storage. *S. sclerotiorum* is a ubiquitous necrotrophic pathogen that attacks a wide range of cultivated and wild plant species. It results in damage of the plant tissue, followed by cell death and soft rot or white mold of the crop [23]. A heavy infection of white mold is of serious concern to both growers and processors because, besides reducing yield, it seriously disrupts the processing operation [24]. The white mold disease is a perpetual problem to bean crops in Kashmir, and its pathogen is soil, seed, and air-borne in nature. Being a short duration crop of 40–45 days, the bean crop is cultivated 3–4 times in a year on the same field which facilitates the buildup of a pathogenic inoculum sufficient to develop disease [22,23,24,25]. In the Kashmir valley, no systemic study on the variability of pathogenic isolates and screening of bean germplasm against white mold disease has been made so far. In the North Kashmir region, this disease is observed to show higher intensity in a few years [12]. The study of the variability of this pathogen is the basic requirement for developing the control measures, especially for the development of resistant lines. Furthermore, an analysis of structure and population dynamics is an essential part of understanding how the underlying mechanisms are involved in the history of this pathogen and its distribution across different geographic areas. Hence, in this study, an attempt was made to understand the variability of the pathogen and look for a resistance source against this disease.

The study was confined to the North Kashmir region only as the disease is more prevalent in this region compared to other regions of the Kashmir valley. It has been observed to cause heavy losses to the crop in this region as these areas are low lying. The fungal pathogen, isolated from the diseased plants, was identified as *Sclerotinia sclerotiorum* (Lib.) de Bary. The mycelium of the isolated pathogen was hyaline, septate, and irregularly branched. The fungus produced sclerotia in cultures as well as in/on inoculated bean plants. The cultural and morphological characteristics of the pathogen were identical to the findings of other researchers [26,27]. The pathogenicity test revealed development of peculiar symptoms akin to white mold disease. The water-soaked lesion/rotting, white cottony mycelial growth, sclerotia formation, and wilting symptoms were identical to the earlier reports [28].

Many laboratories routinely classify the isolates into mycelial compatibility groups as a quick marker for genotyping *S. sclerotiorum* within populations. Assemblage of 80 isolates into 22 MCGs based on their compatibility test suggests a heterogeneous mixture of genotypes of this pathogen in North Kashmir and hence displays high levels of variability. This finding agrees with prior reports on *S. sclerotiorum* MCG population structures on diverse crops [20,29,30,31,32,33,34]. While working on variability of *S. sclerotiorum* isolates on beans, 12 MCGs were formed by 18 isolates from Brazil [34].

The MCGs mostly consisted of the isolates from geographically adjacent areas; however, in some cases, MCGs were observed to be spread in geographically separated areas as well. The incidence of single mycelial compatibility groups from different districts of North Kashmir specifies the existence of identical parental pathogen lineages across this region. This might be owing to the population of this region exchanging beans (seed and pulse) afflicted with resting stages (sclerotia) of this fungus. The other reason may be sclerotia and ascospores’ dispersal through water and/or wind and also sclerotia dispersal through irrigation water, as these districts share a good connectivity through canals, rivers, and irrigation channels. The current findings are supported by the findings of various workers, for example, recurring recovery of MCGs of *S. sclerotiorum* from canola samples made in Ontario in 1989 [35], eastern Ontario and Quebec in 1999, western Ontario in 2000 [32], western Canada in 1990, 1991, and 1992 [36,37,38], and, likewise, in the studies of Anderson and Kohn in 1995, numerous MCGs’ isolates were disseminated on huge geographic areas, with isolate 2 repeatedly isolated across 2000 km (recovered from Manitoba, Ontario, Alberta, Saskatchewan, and subsequently from cabbage in New York) over a 4-year period [39].

Extensive variability studies were conducted for the isolates in terms of their cultural, morphological, and pathogenic characteristics. Differences were observed in cultural characteristics such as colony color, colony type, and mycelial diameter after 24 and 48 HrI. Among the eighty isolates, the majority showed a white colony color, and some of the isolates had off-white colonies. The off-white colonies varied in the intensity of pigmentation but were all classified under the off-white color because there was no marked difference among the isolates, i.e., the same isolate showed different intensity pigmentation in different replications. These results are in confirmation with other researchers, who also found a similar color pattern while working on 14 isolates of *S. Sclerotiorum* [40,41]. While studying variability among different isolates of *S. sclerotiorum* from different hosts, variations in white and brown colors with a predominant white color were observed, but, in the present study, no brown coloration was observed [42]. This might be due to the fact that, in the present study, isolates were collected from a single bean host, while, in their study, isolates were collected from different hosts, thereby indicating the variation in colony color among isolates from different hosts. During the present study, the isolates were observed to form three types of colonies including smooth, fluffy, and fluffy-at-center-only. Similarly, while working on the morphological variability of *S. sclerotiorum* on oilseed Brassica, three types of colonies of scattered, smooth, and fluffy were observed [41], while fluffy, sparse, and compact colony growth were observed in isolates of *S. sclerotiorum* from oilseed and wheat [40,43].

Significant variation was observed in terms of the colony diameter which ranged from 15.33 mm to 29.33 mm after 24 HrI and between 39.33 mm and 69.67 mm after 48 HrI. Our observations are in agreement with the findings of other scientists who, while working on *S. sclerotiorum* in oilseed crops, found significant variation among the isolates in terms of the colony diameter after 24 and 48 HrI [44,45]. Similarly, a significant difference among the *S. sclerotiorum* isolates in the colony diameter was observed from Indian mustard across different regions of North East India ranging from 13 mm to 27 mm after 24 HrI [40].

The study of morphological characteristics such as the number of sclerotia formed at 15 DAI, the size of sclerotia, and the weight of sclerotia revealed significant variation among the isolates. The number of sclerotia per plate at 15 DAI ranged from 14 to 51. Similar variations in the number of sclerotia observed in 14 isolates of *S. sclerotiorum* on rapeseed ranging from 9 in the BWL isolate to 32 in the HSR isolate were observed [40]. Similarly, in other studies on the variability of *S. sclerotiorum* on beans in Brazil, the number of sclerotia was observed to vary from 11 ± 3 (Ss-188) to 30 ± 16 (Ss-23) [46].

The length and width of sclerotia ranged from 2.98 mm to 10.86 mm and 1.42mm to 4.65 mm, respectively, and showed significant variation among the isolates. The variation in sclerotial size has also been reported by other researchers while working on different crops [40,41,46,47,48]. The fresh weight and dry weight also varied significantly between isolates. The fresh weight ranged from 0.51 g to 1.43 g. Similarly, the dry weight of sclerotia per plate ranged from 0.17 g to 0.51 g [40,46]. While working on 118 Brazilian isolates of *S. sclerotiorum* causing white mold of beans, it was observed that there was high cultural and morphological variation in parameters such as colony diameter, color, type, and sclerotial production [46]. This cultural and morphological variability might be due to different environmental conditions which influence vegetative and reproductive phases of the pathogen and the presence of mycovirus resulting in hypovirulent strains of the host fungus showing reduced or delayed morphological characteristics. Similar results were observed on white mold of the French bean, and variability in cultural and morphological parameters was reported to have arisen due to environmental variation and mycoviruses [23]. The variation is also known to arise by sexual reproduction which is one of the major and most important sources of variability generation in organisms. In this study, it was confirmed that, in Kashmir, sexual reproduction of *S. sclerotiorum* does take place and hence represents a reason for variability within and among the different MCGs. Similarly, observations were also made by scientists who reported that sexual reproduction is considered as an important assumption for divergence among the population from different geographical regions [49]. The existence of variability in cultural and morphological characteristics might also be due to higher genetic variability among the isolates as reported by Goswami et al. while working on variations in different isolates of *Rhizoctonia solani* [50].

The six isolates taken from six different MCGs and inoculated on eighteen bean genotypes showed variation in pathogenicity, with isolate G04 showing the highest virulence followed by the G11 isolate, and the least virulent among them was B01. Variation in virulence among the isolates of *S. sclerotiorum* from different locations may be due to the varying ability of isolates to secrete oxalic acid and the release of some enzymes to macerate the plant cell wall and tissue for infection [23]. Differences in pathogenicity among *S. sclerotiorum* isolates have also been reported by other workers [51,52]. Many lines were showing resistance to one or a few isolates and susceptibility to others. This may be due to the selectivity in pathogenicity of these isolates. A good variation was also found in terms of disease scores among the different genotypes. This could be due to the presence of genes of resistance in these lines against *S. sclerotiorum*. Two genotypes, SR-1 and WB-1402, were showing resistant scores. Variety SR-1 showed the highest disease score of 5.33 (intermediate score) against the G04 isolate with an overall resistant score of 2.89. Similar findings were found on the evaluation of plant introductions of *Phaseolus* spp. for resistance to white mold and the evaluation of fava beans for resistance to sclerotinia stem rot caused by *Sclerotinia trifoliorum,* respectively [53,54]. The virulence of isolates was positively correlated with colony diameter. However, there was no correlation between virulence and number of sclerotia, their size, or weight. The findings are in conformity with the findings of other workers observing the factors affecting virulence and found that hypo-virulent isolates showed reduced mycelial growth [55]. Similar observations were reported by Lehner et al. while working on the screening of the bean germplasm in Brazil [46].

Sixty-three bean genotypes were evaluated for resistance to *S. sclerotiorum* including three cultivated varieties under controlled environmental conditions. The most virulent isolate, i.e., G04, was inoculated using the cut stem method. The results indicated that there was significant variability in resistance to *S. sclerotiorum* in the screened bean genotypes. However, the majority of these genotypes were very susceptible to the disease. Only 14 percent of the cultivars were partially resistant to the most virulent isolate, and only one genotype, WB-1402, was resistant. It was previously reported that only 6.5 percent of dry bean cultivars of the evaluated cultivars were partially resistant, whereas more than 15 percent of the fava bean cultivars were resistant to *S. trifoliorum* [53,54].

## 5. Conclusions

White mold disease of the bean is one of the emerging pathogenic threats to the bean crop in Kashmir, especially in the low-lying areas, and has been noticed to be regularly increasing in severity for the last few years. Cultural and morphological variability of isolates of *S. sclerotiorum* from different geographical locations evaluated in in vitro conditions revealed high diversity among the isolates of *S. sclerotiorum* from different districts of North Kashmir based on their mycelial growth, colony characteristics, and sclerotial formations. It was also observed that variation in cultural and morphological characteristics among the isolates of *S. sclerotiorum* occurs within as well as among the districts. MCG analysis showed 22 groups with the MCGs formed mostly by the isolates within the districts. The population structure revealed a mixture of clones at a particular location. Variability studies within and among MCGs revealed that the isolates are less diverse within a MCG, and diversity among MCGs is higher. Pathogenic variability of isolates of *S. sclerotiorum* from different geographical locations and MCGs were investigated under pot conditions. It showed considerable variation in virulence among the isolates. the study of the carpogenic germination of sclerotia revealed that ascospores are formed under natural conditions in Kashmir as well, and they serve as a source of primary inoculum to spread the white mold disease in spring which is reported from Kashmir for the first time. Evaluation of the bean germplasm for resistance against white mold disease revealed that the majority of the screened genotypes were susceptible, while few showed partial resistance (intermediate scores), and only a single genotype, WB-1402, was shown to be resistant to the most virulent isolate (G04). Among three released varieties, Shalimar Rajmash-1 (SR-1) showed an intermediate reaction, while the other two, Arka Anoop and Shalimar French Bean-1 (SFB-1), were susceptible. Hence, in light of the present investigations, it is concluded that the population of the pathogen of the white mold disease of the bean is highly diverse. In the adjacent areas, similar MCGs are found, while, with increased separation, the isolates form new MCGs, and, in a single isolate, more than one MCG can be found. Isolates with higher colony diameters are more virulent than the isolates with lower colony diameters. Moreover, most of the germplasms evaluated for resistance against white mold disease succumb to it, while a few are moderately resistant, and only one genotype, WB-1402, is resistant.

## Figures and Tables

**Figure 1 jof-08-00755-f001:**
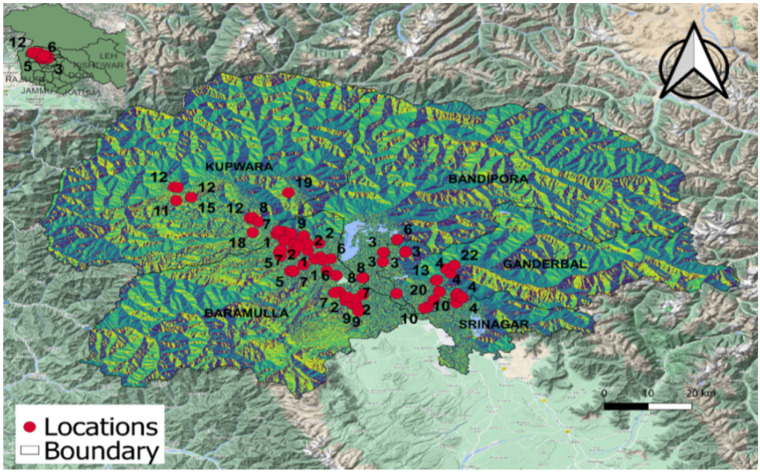
Study area map depicting distribution pattern of isolates of different Mycelial Compatibility Groups.

**Figure 2 jof-08-00755-f002:**
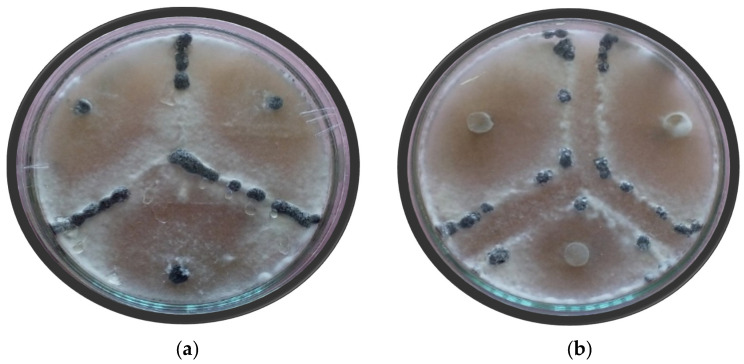
Compatible and incompatible reaction among different isolates of *S. sclerotiorum* from common beans in Kashmir: (**a**) Sclerotia produced at the ridge by compatible isolates; (**b**) clear barrage zone between the incompatible isolates.

**Figure 3 jof-08-00755-f003:**
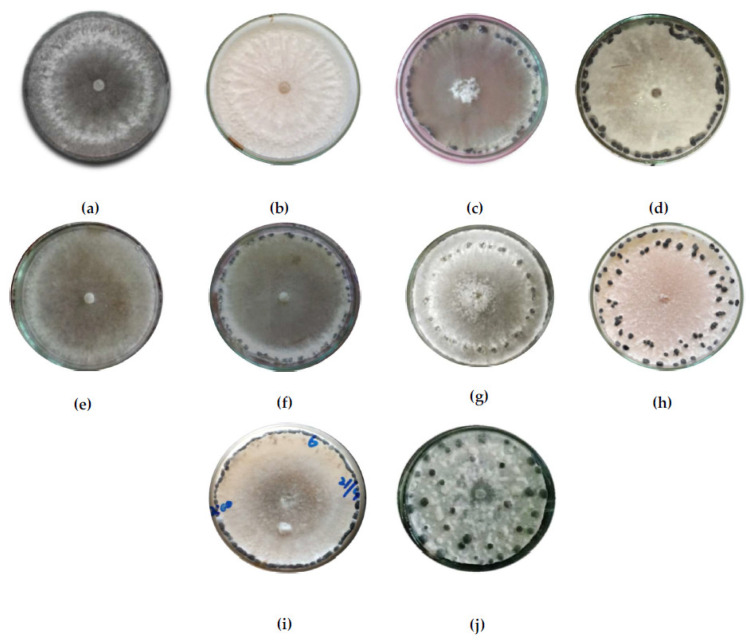
Morph-cultural characteristics: (**a**–**e**) Colony color and type; (**f**–**j**) Different sclerotia formation patterns on PDA of *S. sclerotiorum* isolates. (**a**) off white, fluffy; (**b**) white, fluffy; (**c**) off white, fluffy center; (**d**) white, smooth; (**e**) off white, smooth; (**f**) peripheral; (**g**) sub-peripheral; (**h**) peripheral and sub-peripheral; (**i**) attached to rim; (**j**) scattered.

**Figure 4 jof-08-00755-f004:**
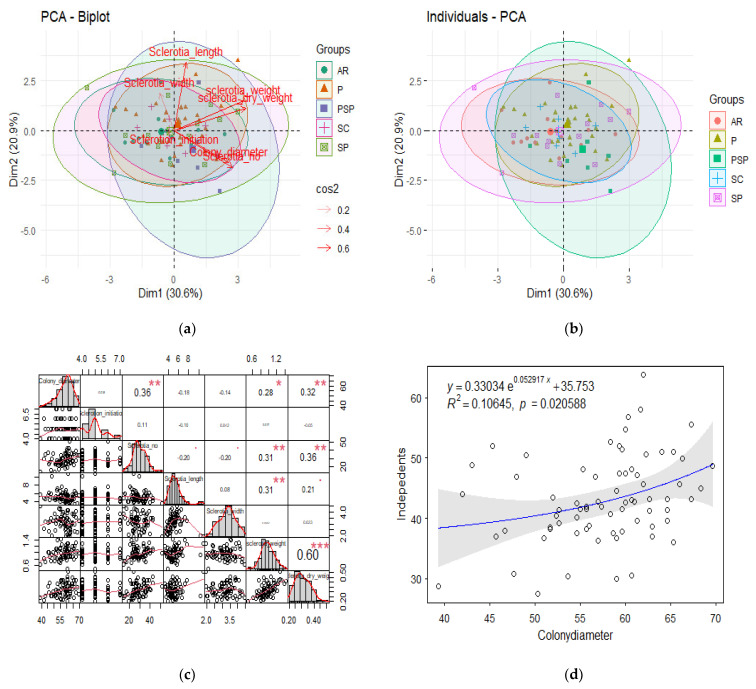
(**a**) Individual biplot among different groups; (**b**) PCA biplot showing orthogonality of variables; (**c**) Correlation relationship between isolates parameters, Here star represents * (*p* ≤ 0.05), ** (*p* ≤ 0.01) and *** (*p* ≤ 0.001); (**d**) Modeling between colony diameter(mm) and independent (size, weight, initial diameter).

**Figure 5 jof-08-00755-f005:**
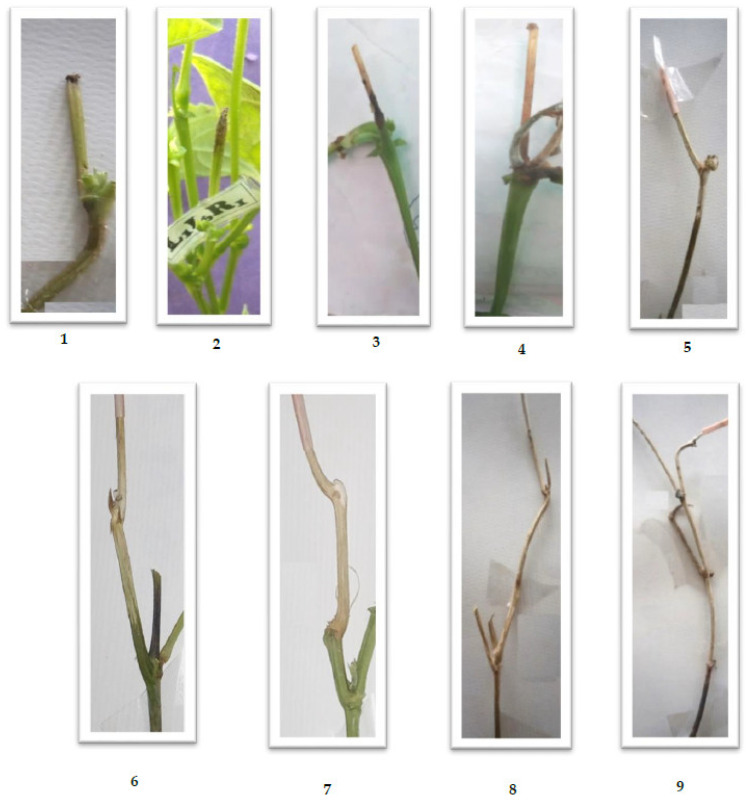
Disease scoring scale; 1–9 represents respective disease scores (Teran et al., 2006) [21].

**Figure 6 jof-08-00755-f006:**
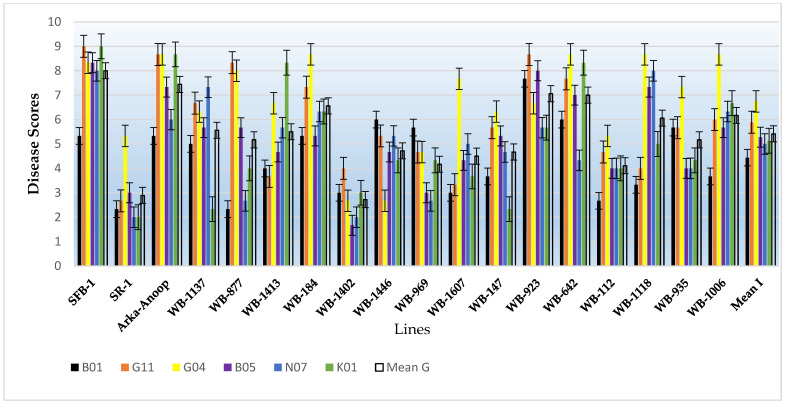
Graphical representation of pathogenicity of six isolates on eighteen bean lines. Mean I = mean disease scores by isolates. Mean G = mean disease scores on common bean lines.

**Figure 7 jof-08-00755-f007:**
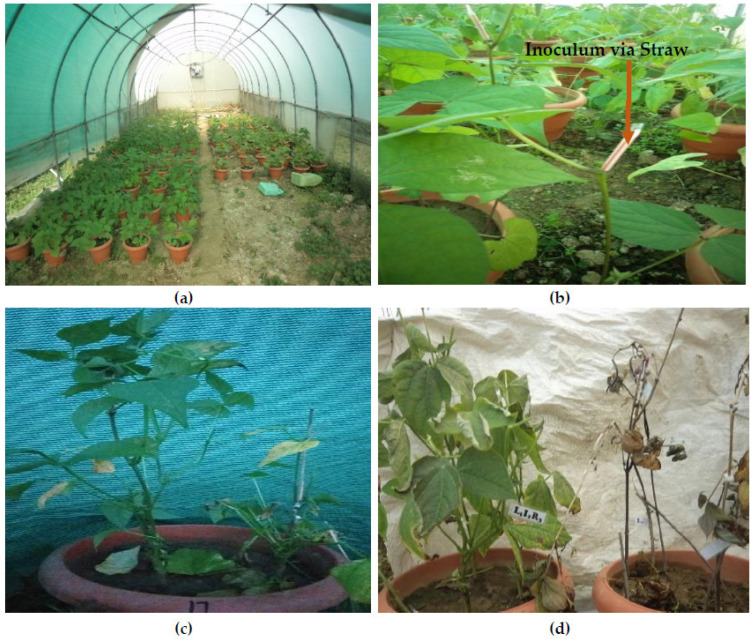
Germplasm screening under controlled conditions against most virulent isolate of mold pathogen: (**a**) General view of bean germplasm before inoculation; (**b**) Inoculated plants; (**c**) Control vs. inoculated plants 15 DAI; (**d**) Resistant line WB-1402 vs. Susceptible line.

**Figure 8 jof-08-00755-f008:**
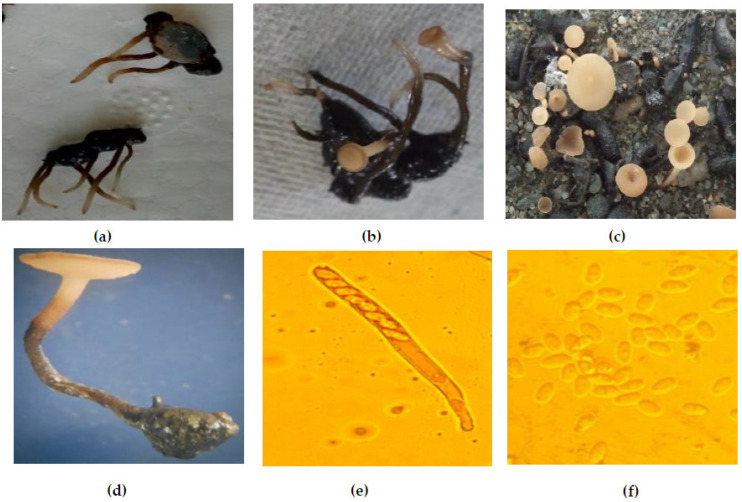
Carpogenic germination of sclerotia of *S. sclerotiorum* isolates from Kashmir. (**a**) Stipe formation; (**b**) developing apothecia; (**c**) overview of apothecia; (**d**) magnified apothecium; (**e**) ascus with ascospores at 100×; (**f**) ascospores at 400×.

**Table 1 jof-08-00755-t001:** The pathogenicity assessment of green mold pathogen against beans plant using resistance scale of Teran et al., 2006.

Grade Value	Symptoms	Scores
1	No sign of branch or stem infection nearby agar inoculant when pipette or straw tube is removed for examination.	Resistant (scores 1–3)
2	Branch or stem infected but invasion of the first internode less than one inch.	
3	Branch or stem invasion of the first internode greater than one inch but not reaching the first node.	
4	Branch or stem invasion reached the first node, but no further.	Intermediate (scores 4–6)
5	Branch or stem invasion passed the first node, but invasion of the second internode less than one inch.	
6	Branch or stem invasion of the second internode greater than one inch but not reaching the second node.	
7	Branch or stem invasion reached the second node, but no further.	Susceptible (scores 7–9)
8	Branch or stem invasion passed the second node, but invasion of the third internode less than one inch.	
9	Branch or stem of the third internode greater than one inch leading to plant death.	

**Table 2 jof-08-00755-t002:** Collection sites with topographic data of 80 Isolates of *S. sclerotiorum* collected from bean plants in North Kashmir.

Isolate	Area	Latitude	Longitude	Altitude	MCG	Isolate	Area	Latitude	Longitude	Altitude	MCG
B01	Wadura	34.354787	74.401002	1583	1	K05	Kulangam	34.41755	74.30565	1595	8
B02	Wadura	34.342693	74.406317	1584	2	K06	Tarathpora	34.46505	74.11725	1845	11
B03	Wadura	34.348281	74.426242	1584	9	K07	Sonwani	34.35276	74.38231	1585	1
B04	Watragam	34.319373	74.370092	1594	5	K08	Sonwani	34.35276	74.38231	1585	1
B05	Watragam	34.308799	74.373922	1593	5	K09	Vilgam	34.47396	74.15442	1772	12
B06	Nathipora	34.366228	74.388203	1588	2	K10	Vilgam	34.47396	74.15442	1772	15
B07	Nathipora	34.377691	74.383089	1588	14	K11	Langate	34.37533	74.30486	1601	18
B08	Nawpora	34.284241	74.42971	1581	1	K12	Unisoo	34.37649	74.3673	1589	1
B09	Nawpora	34.300895	74.457185	1585	14	K13	Unisoo	34.37885	74.37184	1592	15
B10	Duroo	34.34556	74.46819	1587	2	K14	Unisoo	34.37962	74.36803	1589	11
B11	Bomai	34.365888	74.430499	1588	16	K15	Kachloo	34.32716	74.37456	1590	7
B12	Tujjer	34.372279	74.399789	1586	9	K16	Chougal	34.40552	74.31927	1592	1
B13	Doabgah	34.266878	74.397221	1590	5	K17	Kralpora	34.50307	74.11202	1797	12
B14	Doabgah	34.265126	74.405059	1581	7	K18	Kralpora	34.50166	74.12071	1786	11
B15	Doabgah	34.26788	74.405057	1583	5	K19	Sogam	34.4873	74.39276	1717	19
B16	Brath	34.324998	74.440373	1587	2	G01	Saloora	34.21042	74.76315	1587	10
B17	Brath	34.343771	74.438442	1586	1	G02	Saloora	34.20876	74.76399	1585	3
B18	Tarzoo	34.254479	74.510398	1580	6	G03	Batvena	34.24068	74.75522	1596	13
B19	Tarzoo	34.267965	74.487395	1583	2	G04	Batvena	34.24201	74.75782	1597	4
B20	Adipora	34.301649	74.496693	1579	6	G05	Shallabug	34.16641	74.73526	1583	13
B21	Armpora	34.306671	74.471447	1585	1	G06	Shallabug	34.16282	74.72700	1581	10
B22	Armpora	34.297237	74.477099	1583	2	G07	Hakimgund	34.18533	74.74999	1583	20
B23	Tapper	34.18442	74.533395	1593	6	G08	Nunner	34.27329	74.77995	1650	4
B24	Churru	34.208829	74.516692	1584	8	G09	Nunner	34.2609	74.79123	1640	4
B25	Churru	34.206367	74.506237	1593	7	G10	Kujar	34.1832	74.80623	1601	4
B26	Lollipora	34.201365	74.575666	1578	8	G11	Kujar	34.17761	74.80569	1584	3
B27	Palhallan	34.17765	74.558118	1581	7	G12	Nagbal	34.2039	74.80783	1620	10
B28	Palhallan	34.174019	74.551236	1582	1	G13	Nagbal	34.19864	74.80574	1612	10
B29	Palhallan	34.188175	74.558509	1583	2	G14	Nagbal	34.20208	74.80672	1618	21
B30	Tapper	34.194657	74.53278	1592	2	G15	Manigam	34.2833	74.80048	1693	22
B31	Tapper	34.194254	74.538117	1589	17	G16	Shuhama	34.19161	74.821	1618	4
B32	Pattan	34.163097	74.562978	1582	9	G17	Shuhama	34.19467	74.81906	1622	4
B33	Pattan	34.153722	74.565663	1585	9	N01	Shilwat	34.20363	74.65854	1584	13
B34	HibDangerpora	34.319858	74.407793	1586	5	N02	Naidkhai	34.24812	74.57697	1584	8
B35	HibDangerpora	34.322131	74.404928	1584	1	N03	Naidkhai	34.24756	74.57481	1583	8
B36	Behrampora	34.336087	74.418171	1583	6	N04	Prang	34.29328	74.62488	1581	3
K01	Ujroo	34.369972	74.366456	1589	7	N05	Bazipora	34.32225	74.68102	1583	3
K02	Ujroo	34.37128	74.363458	1588	1	N06	Bazipora	34.31975	74.68034	1585	3
K03	Kulangam	34.415695	74.298381	1594	12	N07	Saderkoot	34.35588	74.66041	1579	6
K04	Kulangam	34.412252	74.306513	1592	11	N08	Madwan	34.31977	74.62591	1580	3

**Table 3 jof-08-00755-t003:** Mycelial compatibility groups of *S. Sclerotiorum* pathogen on beans in Kashmir, constituent isolates, and area of occurrence.

MCG	No. of Isolates	Isolate Codes	Area of Occurrence
01	11	B28, B35, K02, K07, B01, B08, B17, B21, K08, K12, K16	Kupwara and Baramulla
02	08	B10, B16, B02, B06, B19, B30 B22, B29	Baramulla
03	06	G02, G11, N04, N05, N06, N08	Ganderbal and Bandipora
04	06	G10, G16, G04, G08, G09, G17	Ganderbal
05	05	B04, B15, B05, B13, B34	Baramulla
06	05	B36, B18, B20, B23, N07	Baramulla and Bandipora
07	05	B27, K01, B14, B25, K15	Baramulla and Kupwara
08	05	N02, B24, B26, K05, N03	Baramulla, Bandipora and Kupwara
09	04	B32, B33 B03, B12	Baramulla
10	04	G01, G12,G06, G13	Ganderbal
11	04	K06, K14, K04, K18	Kupwara
12	03	K09, K17, K03	Kupwara
13	03	N01, G03, G05	Bandipora and Ganderbal
14	02	B09, B07	Baramulla
15	02	K13, K10	Kupwara

Seven MCGs were represented by a single isolate each, specifically, G07, G15, B11, B31, G14, K11, and K19.

**Table 4 jof-08-00755-t004:** Morpho-cultural characteristics of 80 isolates of *S. sclerotiorum* isolates from bean crops collected from North Kashmir.

Isolate	Colony Characteristics					Sclerotial Parameters					
	Color	Type	Diameter (mm)		Initiation DAI	No./plate	Size (mm)		Weight (g)		Pattern
			24 h	48 h			Length	Width	Fresh	Dry	
B01	White	Smooth	25.33	60.67	4	33.67	5.28	3.89	1.29	0.43	P
B02	White	Fluffy	21.33	52.33	5	24.33	6.7	3.29	1.11	0.24	P
B03	White	Smooth	24.00	59.33	5	37.67	5.3	2.92	1.15	0.33	P
B04	White	Fluffy center	26.00	62.00	6	51.33	3.5	1.97	0.98	0.36	P and SP
B05	White	Smooth	27.67	65.33	4	36.67	5.4	3.82	1.12	0.42	SP
B06	Off white	Smooth	17.33	45.67	4	23.33	6.21	2.41	1.02	0.29	SC
B07	White	Smooth	16.33	43.00	5	35.33	4.6	3.17	0.7	0.26	AR
B08	White	Fluffy	25.33	60.00	5	28.67	5.31	2.84	0.76	0.31	P
B09	White	Smooth	17.33	45.33	5	37.33	5.51	3.14	1.02	0.36	P
B10	White	Smooth	22.33	54.67	6	22	5.49	3.34	0.68	0.23	SP
B11	Whiten	Fluffy	23.33	56.00	6	20.67	7.6	3.28	1.25	0.42	P and SP
B12	White	Fluffy	19.33	49.00	5	36.33	4.87	3.27	1	0.3	P
B13	White	Smooth	27.33	62.67	5	28.67	4.42	2.33	0.86	0.29	AR
B14	White	Smooth	24.00	58.67	4	23.33	6.27	2.95	0.98	0.38	P
B15	Off white	Fluffy	25.33	60.00	5	34	4.5	2.98	1	0.39	P and SP
B16	White	Smooth	22.00	55.67	6	23	5.34	2.97	0.85	0.22	AR
B17	White	Fluffy	23.00	57.33	5	26.33	6.17	3.51	0.94	0.42	P
B18	White	Smooth	22.67	54.67	6	24.33	4.77	4.3	0.78	0.27	SC
B19	White	Fluffy	19.33	52.67	5	22	7.26	3.55	1.39	0.38	P
B20	Off white	Smooth	25.67	61.00	5	23.67	10.86	1.96	1.28	0.51	P
B21	White	Smooth	23.33	61.33	5	32.67	4.53	4.14	0.86	0.25	SP
B22	White	Fluffy	20.33	51.67	5	23.67	5.18	3.32	1.11	0.32	P
B23	Off white	Fluffy center	20.00	52.00	7	27.67	4.65	3.26	0.82	0.29	P
B24	Off white	Fluffy	21.33	52.67	6	25.67	5.12	3.56	1.06	0.31	P
B25	White	Smooth	22.33	55.00	5	28.67	3.9	2.83	1.15	0.37	SP
B26	White	Fluffy center	25.00	61.33	4	26	5.5	3.14	1.02	0.36	P
B27	White	Fluffy center	23.00	59.67	5	23.67	5.55	2.35	1.22	0.38	P
B28	White	Smooth	26.00	67.33	6	27	5.72	3.56	0.87	0.29	AR
B29	White	Smooth	22.67	55.67	5	26.67	4.86	3.53	1.43	0.41	SP
B30	White	Smooth	18.00	46.67	4	25.67	4.31	3.16	0.87	0.39	AR
B31	White	Smooth	22.33	60.67	5	19.00	3.95	2.09	0.48	0.20	SP
B32	White	Smooth	24.67	60.00	4	40.67	5.5	3.45	1.07	0.32	AR
B33	White	Fluffy	25.00	61.67	7	42.33	4.27	3.3	1.13	0.39	P and SP
B34	White	Fluffy	24.33	59.00	5	32	4.13	2.52	0.88	0.32	P
B35	White	Smooth	22.00	57.00	4	22	5.54	3.95	0.87	0.22	P
B36	White	Fluffy	25.00	60.67	5	27	5.7	4.12	1.21	0.31	P
N01	White	Smooth	18.00	48.00	4	29.67	10.25	2.03	0.94	0.26	AR
N02	White	Fluffy	26.33	63.00	4	25	4.43	2.61	0.91	0.29	SC
N03	White	Fluffy	23.33	55.67	5	27	5.58	3.41	0.99	0.32	SC
N04	White	Smooth	27.33	66.00	5	33.67	3.36	2.42	1.2	0.38	P
N05	White	Smooth	27.67	66.33	5	34.33	6.53	2.96	1.04	0.35	P
N06	White	Smooth	26.67	64.00	6	36.33	4.67	2.97	1.06	0.36	P
N07	White	Smooth	24.67	60.33	5	42.33	4.88	3.51	1.06	0.46	AR
N08	White	Smooth	26.33	62.00	5	33.33	3.66	2.7	0.97	0.25	P
G01	White	Smooth	24.67	59.00	5	24	3.74	3.16	0.87	0.31	SP
G02	White	Smooth	28.00	67.33	5	39	6.19	4.06	1.31	0.42	SP
G03	White	Smooth	27.00	62.33	4	39.67	3.77	2.14	0.97	0.33	P and SP
G04	White	Smooth	29.00	68.33	4	34	3.53	2.35	1.03	0.35	SP
G05	Off white	Fluffy center	25.33	58.33	6	36	5.84	3.66	1.13	0.34	SP
G06	White	Smooth	20.33	51.00	6	23	4.43	2.43	0.85	0.27	P and SP
G07	White	Smooth	20.33	53.67	4	16.67	4.81	4.25	0.70	0.30	P
G08	Off white	Smooth	15.33	39.33	4	14.00	5.70	4.45	0.62	0.24	SP
G09	White	Smooth	24.67	63.00	7	24.33	3.85	2.51	1.00	0.28	SP
G10	White	Smooth	24.33	64.67	5	26.33	4.31	3.02	0.97	0.33	P
G11	White	Smooth	29.33	69.67	5	34.00	5.46	3.28	0.99	0.28	AR
G12	White	Smooth	25.00	59.00	4	18.00	3.99	3.07	0.84	0.28	P
G13	Off white	Smooth	18.67	50.33	4	14.00	4.86	3.85	0.73	0.26	AR
G14	White	Smooth	22.67	55.33	5	34.33	4.64	3.34	0.75	0.42	P
G15	White	Smooth	25.67	65.33	5	22.00	4.56	3.7	0.79	0.24	AR
G16	White	Smooth	20.67	51.67	7	22.67	4.85	3.06	0.97	0.27	SP
G17	Off white	Smooth	17.67	47.67	5	15.67	5.92	3.48	0.73	0.26	P
K01	White	Smooth	26.00	62.67	4	31.67	4.38	2.26	0.90	0.34	SC
K02	White	Smooth	24.00	58.33	4	27.67	5.06	2.92	1.06	0.25	AR
K03	Off white	Fluffy	22.00	54.67	5	25.67	6.52	4.17	1.13	0.29	SP
K04	White	Smooth	23.67	59.33	5	35.00	3.87	2.63	0.93	0.30	P and SP
K05	White	Smooth	26.67	64.67	5	27.00	5.14	3.08	1.19	0.38	SC
K06	Off white	Smooth	24.33	59.67	4	30.33	4.32	2.01	0.83	0.26	P
K07	Off white	Smooth	23.33	57.00	5	29.33	4.26	3.41	0.82	0.37	SC
K08	White	Smooth	23.67	64.67	5	27.67	5.77	3.85	0.88	0.45	P
K09	White	Smooth	15.67	42.00	6	29.67	4.89	2.64	0.81	0.21	AR
K10	White	Smooth	25.00	59.67	5	35.33	6.88	3.31	0.91	0.30	P
K11	Off white	Smooth	24.33	56.67	7	30.67	4.53	3.8	0.87	0.34	P
K12	White	Smooth	23.33	59.67	5	25.00	4.45	2.81	0.99	0.31	SP
K13	White	Smooth	22.67	55.33	5	40.67	5.16	2.87	1.11	0.46	P and SP
K14	White	Smooth	25.00	62.00	5	37.33	10.10	2.86	1.15	0.38	P
K15	Off white	Fluffy	24.33	61.33	4	45.33	7.02	2.12	1.40	0.35	P and SP
K16	Off white	Smooth	20.00	52.33	4	23.67	4.90	3.03	0.74	0.28	P
K17	White	Smooth	21.00	53.00	5	17.67	4.45	1.42	0.51	0.17	P
K18	White	Smooth	22.67	59.00	6	36.33	3.79	1.69	0.92	0.32	SP
K19	White	Smooth	17.33	45.67	7	19.33	2.98	2.89	0.85	0.25	P
C.D. (*p* ≤ 0.05)			2.37	4.78		9.27	2.45	1.05	0.41	0.14	

P = Peripheral ring, SP = Sub Peripheral ring, P and SP = Peripheral and sub peripheral rings, AR = Arranged at Rim, SC = Scattered.

**Table 5 jof-08-00755-t005:** Morpho-cultural variability between 22 MCGs formed by 80 *S. sclerotiorum* isolates collected from beans.

MCG	Diameter 24 HrI	Diameter 48 HrI	No. of Sclerotia per Plate	Fresh Weight	Dry Weight	Length	Width
1	23.85 ± 0.62	60.51 ± 0.87	27.91 ± 0.78	0.96 ± 0.07	0.32 ± 0.03	5.05 ± 0.10	3.38 ± 0.29
2	20.46 ± 0.21	52.09 ± 0.50	24.92 ± 1.75	1.00 ± 0.07	0.31 ± 0.03	5.83 ± 0.02	3.19 ± 0.24
3	27.22 ± 0.13	66.06 ± 0.90	33.77 ± 1.20	1.09 ± 0.09	0.34 ± 0.02	4.95 ± 0.04	2.83 ± 0.26
4	21.94 ± 0.87	55.78 ± 1.32	22.83 ± 1.32	0.88 ± 0.07	0.28 ± 0.02	4.69 ± 0.07	3.18 ± 0.19
5	25.73 ± 0.35	61.6 ± 1.40	36.53 ± 3.34	0.97 ± 0.07	0.35 ± 0.03	4.39 ± 0.06	2.64 ± 0.22
6	23.4 ± 0.50	57.73 ± 1.51	29.00 ± 0.23	1.03 ± 0.08	0.36 ± 0.01	6.16 ± 0.17	3.39 ± 0.13
7	23.93 ± 0.28	59.47 ± 0.22	30.53 ± 0.47	1.13 ± 0.09	0.37 ± 0.02	5.42 ± 0.04	2.50 ± 0.21
8	24.53 ± 1.52	59.47 ± 2.76	26.13 ± 0.52	1.03 ± 0.08	0.33 ± 0.02	5.15 ± 0.08	3.16 ± 0.29
9	21.58 ± 1.71	56.25 ± 2.16	33.42 ± 1.84	0.92 ± 0.06	0.29 ± 0.02	4.84 ± 0.06	2.86 ± 0.24
10	20.17 ± 1.04	50.83 ± 2.51	19.75 ± 0.32	0.82 ± 0.05	0.28 ± 0.03	4.25 ± 0.09	3.13 ± 0.33
11	24.17 ± 0.78	60.25 ± 1.41	34.75 ± 1.01	0.95 ± 0.07	0.31 ± 0.03	5.52 ± 0.30	2.05 ± 0.12
12	19.56 ± 0.57	49.89 ± 1.27	24.34 ± 4.55	0.82 ± 0.06	0.22 ± 0.04	5.29 ± 0.50	2.74 ± 0.27
13	23.44 ± 0.97	56.22 ± 1.60	35.11 ± 2.16	1.01 ± 0.08	0.31 ± 0.03	6.62 ± 0.80	2.54 ± 0.28
14	16.83 ± 1.35	44.16 ± 2.67	36.33 ± 2.57	0.86 ± 0.09	0.31 ± 0.01	4.96 ± 0.79	3.10 ± 0.32
15	23.33 ± 0.14	56.5 ± 0.56	35.50 ± 3.22	1.01 ± 0.08	0.38 ± 0.02	5.82 ± 0.63	2.89 ± 0.37
16	23.33 ± 1.45	56 ± 4.03	20.67 ± 1.20	1.25 ± 0.10	0.42 ± 0.01	7.60 ± 1.45	3.28 ± 0.04
17	22.33 ± 1.20	60.67 ± 3.51	19.00 ± 2.33	0.48 ± 0.05	0.20 ± 0.04	3.95 ± 0.53	2.09 ± 0.23
18	20.33 ± 0.87	53.67 ± 2.83	16.67 ± 2.31	0.70 ± 0.03	0.30 ± 0.02	4.81 ± 0.26	4.25 ± 0.02
19	22.67 ± 2.02	55.33 ± 2.95	34.33 ± 1.20	0.75 ± 0.06	0.42 ± 0.03	4.64 ± 0.38	3.34 ± 0.26
20	25.67 ± 1.44	65.33 ± 3.17	22.00 ± 1.53	0.79 ± 0.06	0.24 ± 0.03	4.56 ± 0.48	3.70 ± 0.28
21	24.33 ± 1.44	56.67 ± 2.39	30.67 ± 2.03	0.87 ± 0.07	0.34 ± 0.03	4.53 ± 0.83	3.80 ± 0.61
22	17.33 ± 1.19	45.67 ± 1.83	19.33 ± 1.45	0.85 ± 0.11	0.25 ± 0.02	2.98 ± 0.56	2.89 ± 0.24
C.D. (*p* < 0.05)	2.58	4.64	5.71	0.12	0.08	1.28	0.60

**Table 6 jof-08-00755-t006:** Disease reaction of six isolates of *S. sclerotiorum* on eighteen bean genotypes.

	Isolates	B01	G11	G04	B05	N07	K01	Mean
Genotypes								
SFB-1		5.33	9.00	8.33	8.33	8.00	9.00	8.00
SR-1		2.33	2.67	5.33	3.00	2.00	2.00	2.89
Arka-Anoop		5.33	8.67	8.67	7.33	6.00	8.67	7.44
WB-1137		5.00	6.67	6.33	5.67	7.33	2.33	5.56
WB-877		2.33	8.33	8.00	5.67	2.67	4.00	5.17
WB-1413		4.00	3.67	6.67	4.67	5.67	8.33	5.50
WB-184		5.33	7.33	8.67	5.33	6.33	6.33	6.56
WB-1402		3.00	4.00	2.67	1.67	2.00	3.00	2.72
WB-1446		6.00	5.33	2.67	4.67	5.33	4.33	4.72
WB-969		5.67	4.67	4.67	3.00	2.67	4.33	4.17
WB-1607		3.00	3.33	7.67	4.33	5.00	3.67	4.50
WB-147		3.67	5.67	6.33	5.33	4.67	2.33	4.67
WB-923		7.67	8.67	6.67	8.00	5.67	5.67	7.06
WB-642		6.00	7.67	8.67	7.00	4.33	8.33	7.00
WB-112		2.67	4.67	5.33	4.00	4.00	4.00	4.11
WB-1118		3.33	4.00	8.67	7.33	8.00	5.00	6.06
WB-935		5.67	5.67	7.33	4.00	4.00	4.33	5.17
WB-1006		3.67	6.00	8.67	5.67	6.33	6.67	6.17
Mean		4.44	5.89	6.74	5.28	5.00	5.13	
Factors				C.D. (*p* < 0.05)	SE(d)			SE (m)
Lines				0.87	0.443			0.313
Isolates				0.50	0.256			0.181
Lines × Isolates				2.13	1.085			0.767

**Table 7 jof-08-00755-t007:** Screening results of bean genotypes against the most virulent isolate.

S. No.	Line	Reaction	S. No.	Line	Reaction	S. No.	Line	Reaction
01	WB-341	S	22	WB-1319	S	43	WB-54	I
02	WB-112	I	23	WB-1436	S	44	WB-1181	S
03	WB-923	S	24	WB-195	S	45	WB-435	S
04	WB-1413	S	25	WB-1587	S	46	WB-662	S
05	WB-877	S	26	WB-258	S	47	WB-245	S
06	WB-1402	R	27	WB-1634	I	48	WB-969	I
07	WB-935	S	28	WB-1256	S	49	WB-956	S
08	WB-206	S	29	WB-1390	S	50	WB-846	S
09	WB-185	S	31	WB-243	S	51	WB-102	S
10	WB-141	S	30	WB-852	S	52	WB-1185	S
11	WB-448	S	32	WB-352	S	53	WB-435	S
12	WB-184	S	33	SFB-1	S	54	WB-1318	S
13	WB-1607	I	34	WB-371	I	55	WB-966	S
14	WB-1006	I	35	WB-1496	S	56	WB-864	S
15	WB-195	S	36	WB-1129	S	57	WB-6960	S
16	WB-4564	S	37	WB-1441	I	58	WB-1644	S
17	WB-642	S	38	WB-1643	I	59	WB-642	S
18	WB-1137	S	39	SR-1	I	60	WB-1446	I
19	WB-1118	I	40	WB-482	S	61	WB-662	S
20	WB-1304	S	41	WB-1518	S	62	WB-335	S
21	WB-952	S	42	WB-147	S	63	Arka-Anoop	S

## Data Availability

Not Applicable.

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
