# Peer review of "Morpho-Cultural and Pathogenic Variability of Sclerotinia sclerotiorum Causing White Mold of Common Beans in Temperate Climate"

_jof, 2022, doi:10.3390/jof8070755_

Round 1

Reviewer 1 Report

Sclerotinia sclerotiorum is an important necrotrophic fungal pathogen. This manuscript collected eighty isolates of S. sclerotiorum from major common bean production belts of North Kashmir and analyzed the variability of these isolates on cultural, morphological, and pathogenic. The authors also screened the Sclerotinia resistance among the bean germplasm and identified a resistant line. Overall, the authors systemically revealed the population of the S. sclerotiorum in North Kashmir. However, some problems that need to be resolved. Besides, the presentation of whole data in the manuscript can be improved.

1. The authors analysed the cultural and morphological characteristics like colony colour and type, colony diameter, number of days for sclerotia initiation, sclerotia number per plate, sclerotial weight and size among the eighty isolates. The compatibility analysis was conducted to group the eighty isolates into 22 MCGs. The main concern is whether the authors could provide some molecular evidence to distinguish the 22 MCGs or 80 isolates? That will make the results more reliable.

2. Another main concern is the pathogenic variability analysis. Why the authors do not select all 80 isolates for pathogenicity test? Although the selected 6 strains were at the average level in mycelium growth, there are many factors affect the pathogenicity of S. sclerotiorum. I think the authors should conduct an overall evaluation on the pathogenicity of 80 isolates.

3. The statistical analysis used in this manuscript is not clearly clarified. As a matter of fact, the results could be further analysed. ANOVA can be used to compare the means among the MCG groups or among the isolates. In Table 4, Table 5 and Table 6, no comparisons were presented to show the significant difference between isolates or MCGs. Meanwhile, I suggest the data presented in these Tables with means and standard deviation.

4. Some important information seems to be missed in the legend of Figure 4. Neither Figure 4a nor 4b was not presented in the main text. Furthermore, the presentation of Figure 4 needs be improved and clarified more clearly.

5. In the carpogenic germination analysis, the authors mentioned that ‘a lot of variability in terms of MCGs was seen which suggested spread of the inoculum to longer distances through ascospores’. Is there any evidence for the correlating of variability of MCGs with the ascospores spread?

Author Response

Dear reviewer, we are highly thankful for your valuable suggestion and knowledge. The whole manuscript has been thoroughly reviewed for correction/suggestion and has been improved in the revised manuscript.

Author Response

Dear reviewer, we are highly thankful for your valuable suggestion and knowledge. The whole manuscript has been thoroughly reviewed for correction and has been improved in the revised manuscript

Round 2

Reviewer 1 Report

The authors have answered all the questions. I suggest the authors add a figure legend of Figure 4.

This manuscript is a resubmission of an earlier submission. The following is a list of the peer review reports and author responses from that submission.

Round 1

Reviewer 1 Report

This article mainly studies the culture morphology and pathogenicity of Sclerotinia sclerotiorum for selecting the germplasm of bean.  This is a laborious and interesting article. However, the authors seem unclear in the experimental design and analysis process. For example, they used RCBD (line 179) for screening germplasm but used CRD for data analysis (line 196). So, I suggest rejecting the manuscript.

Reviewer 2 Report

Dear Dr

Happy day

The paper titled: Morpho-cultural and Pathogenic Variability of Sclerotinia

sclerotiorum causing White Mold of Beans, contain interesting geographical study for the distribution of the fungus.

I suggest the following points to improve the paper:

1- insert a diagram with the actual images (already existed) to show the life cycle of the fungus during infection.

2- Provide images for the seeds that have been used for the plants cultivation and any survival seeds after the experiments.

3- Describe the environmental factors of the described locations like the humidity, the day length, the temperature, month time of cultivation etc. And did that combatable with the cultivation area.

4- How the authors apply different isolates using the same cultivation area?

5-More information about the cultivation conditions and the fungal growth conditions are required. As well as the distribution of various isolates in the area described in the paper should be clear in the map